# Challenges, Opportunities, and Coping in the Wake of the COVID-19 Pandemic: The Case of Jewish Communities around the World

**DOI:** 10.3390/ijerph20021107

**Published:** 2023-01-08

**Authors:** Orna Braun-Lewensohn

**Affiliations:** Conflict Management & Resolution Program, Ben-Gurion University of the Negev, Beer-Sheva 8410501, Israel; ornabl@bgu.ac.il

**Keywords:** Jewish communities, COVID-19, resiliency, sense of community, coping, health, mental health

## Abstract

Against the backdrop of the COVID-19 pandemic, which lasted more than two years and included several waves, the present study focused on Jewish communities around the world, in order to understand the role of community during the pandemic. This study focused on the community mechanisms that helped community members to cope with the pandemic. To that end, between October 2021 and July 2022, in-person interviews were conducted with leaders and members of the following communities: Budapest, Hungary; Subotica, Serbia; Vienna, Austria; Bratislava, Slovakia; Vilna, Lithuania; Buenos Aires, Rosario, Salta, and Ushuaia in Argentina; and Mexico City and Cancun in Mexico. Each interview lasted between 45 min and 1.5 h. All of the interviews were audio-recorded and transcripts of those recordings were prepared. Three major themes emerged from the interviews: challenges, coping, and opportunities. Most of these themes were common to the different communities around the world. The findings of this work are discussed in terms of the concept of sense of community and resiliency theories.

## 1. Introduction

The global COVID-19 pandemic led the governments of many countries around the world to ask their citizens to minimize gatherings. Educational systems (including schools, kindergartens, colleges, and universities) were closed for a long period of time. In old-age homes, restrictions were enforced and no one was allowed to enter the homes for several months. In addition, there were periods of lockdowns, which left people isolated, anxious, and facing economic difficulties. When vaccinations began to become available throughout the world, some individuals were eager to receive them, while others were afraid and resisted the opportunity to get vaccinated.

Against the backdrop of this pandemic, which lasted more than two years and included several waves, the present study aimed to focus on Jewish communities around the world, in order to understand the role of community during this pandemic. We chose Jewish communities, since minority individuals during periods of stress, including chronic stressful times, seems to be especially vulnerable. Based on the sense of community theory [1,2] and its related concepts, this study examined the ways in which Jewish communities helped their members to cope with the lockdowns and other difficulties that emerged as result of the COVID-19 pandemic, mainly through the eyes of community stakeholders. We further sought to explore challenges and opportunities presented by the pandemic, as noted by the interviewees for their communities. Thus, Jewish communities around the world were chosen to be a case study with a focus on how minority communities with special characteristics used their unique features and qualities to help their members to cope with the pandemic. The literature review summarizes the importance of these special attributes of communities and, at the end of the review, specific goals and questions are formulated to address these issues.

### 1.1. Sense of Community

In recent decades, there has been growing interest in the ideas of community, belonging to a community, and the benefits that communities offer their members. In an era of globalization, it is important to note that community is a construct that crosses cultures and borders. Globalization also amplifies the need for social support and the need for a sense of belonging [3].

Sense of community is a feeling of connection (trust and belonging) with other members of one’s group or residents of one’s locale [4], which includes shared values and concerns. Sense of community is an aspect of community capacity and includes strong concern for community issues, respect for and service to others, sense of connection, and the fulfillment of needs [5]. A strong sense of community among community members contributes to a community’s resilience [6,7,8]. Sense of community is a personal quality connoting individuals’ strong attachments to their communities [2]. McMillan and Chavis [1] defined this concept in terms of four elements: membership, influence, integration and fulfillment of needs, and shared emotional connections. Membership includes sense of belonging and identification, emotional security, the possibility of personal space and personal investment, and a common system of symbols. Influence includes individual interest in a community in which individuals believe that they have some degree of direct and indirect influence, as well as conformity. Integration and fulfillment of needs include dimensions that strengthen the sense of togetherness within a community and the confidence that the needs of individuals will be met by others in the community. Finally, shared emotional connections refer to the level and quality of connection between the different members of the community, who have a shared history, share important experiences, and work together to tackle common problems. Research suggests that these elements are interrelated and together comprise a fairly cohesive construct [2,9,10].

According to the sense-of-community model, people who feel a strong sense of community in relation to a community entity will feel connected to that entity and will see themselves as able to influence the community and to be influenced by it. They will believe that their needs are being met within the community and will feel a sense of obligation to that community [2]. Sense of community is not a result of concrete experience, but rather a mode of thinking in which individuals see themselves as part of a community whose resources are or will be available to them if and when they are needed. Research shows that sense of community can protect against the development of depressive symptoms [11], symptoms of posttraumatic stress, and other emotional problems faced by those who are or have been exposed to stressful events [12].

### 1.2. Community Resilience and Social Capital

Resilience is a “clustered” phenomenon; it occurs among groups of people who have meaningful relationships [13]. Community resilience emphasizes availability of community resources. Adger [14] claimed that community resilience is the quantity and quality of resources available to the community, in addition to those resources that can be applied to meet new challenges. According to Breton [15], community resilience is the human and social capital present within the community. In this context, people, networks, and voluntary associations are the social capital that can effectively move individuals to action. The infrastructure and community services are also considered important social capital. A resilient community can bear internal conflict while preserving the diversity and variety of its individual members and groups [16]. It also provides its members with resources that can help them to cope with adversity.

Adversity can lead a community to develop new modes of functioning. A resilient community can adapt to new conditions and situations by creating new practices and new institutions that uphold its values. It is important to consider community resilience in terms of its effects on individuals and to consider potential inequalities experienced by some groups or individuals within the community.

Social support refers to the material, instrumental, and emotional assistance that individuals receive from others under normal circumstances and in times of crisis. This support is often a function of social networks. Social networks are the essence of community; they comprise relationships between individuals and groups, which involve a variety of emotional and practical elements.

Under ordinary circumstances and at times of special need, social networks can provide economic, material, and informational resources; assist with solving problems; and provide various forms of support. Individuals are embedded in networks, which provide them with social roles and status, as well as personal direction and a common purpose [17].

Community resilience can be a protective factor in the face of violence and disasters [18]. Community resilience includes a community’s preparedness for such events, as well as social support, social ties, and commitments in the community, which are assets under stressful circumstances [19]. Social cohesion is an important part of community resilience [20]. A community’s preparedness and ability to cope well with a crisis depends on the presence of community leaders who can provide authentic and grassroots leadership [21]. These leaders can create community competence with the capacity for collective action and decision-making, which will empower members of the community [22]. Community resilience and wellness emerge from a range of dynamic abilities, which must be continuously cultivated and strengthened.

Most studies have found that community resilience is an important asset for community members and that it facilitates adaptation in stressful situations (e.g., disasters, political violence; ref. [23]), including during the COVID-19 pandemic [24,25]. However, there have been few studies of minority communities around the world during COVID-19 and even fewer have specifically examined Jewish communities around the world during that period. The few studies that have addressed Jewish communities as minority communities during COVID-19 have focused on Orthodox communities, especially in the USA (e.g., [26,27]), and stressed their religious coping [28].

### 1.3. Jewish Communities during COVID-19

To date, six studies have been published on Jewish communities around the world in the wake of the COVID-19 pandemic. Four of these studies were quantitative and focused on Orthodox Jewish communities in the United States [26,27,29,30], one focused on ultra-Orthodox communities in Antwerp, Belgium [28], and one was based on interviews with elderly members of Jewish communities in Europe, Rio de Janeiro, New York, and Israel [31]. Most of these studies examined the roles of religiosity, spirituality, and faith in helping Jewish community members to overcome the stressful situation of the pandemic [26,29,30]. Vanhamel et al. [28] focused on the role of religious leaders in transmitting health messages to their communities, while an oral-history project [31] addressed the importance of the community for its members during a pandemic through the eyes of the elderly.

### 1.4. The Current Study

The concept of community encompasses various theories and dimensions in which previous studies have shown the importance of communities and their characteristics in helping their members during different crises such as political violence and disasters. The different dimensions and concepts related to communities like the sense of community, community resilience, and social capital have been shown to significantly have a protective effect, especially for minorities [32]. Therefore, it was suggested that policymakers, educators, and health practitioners will be aware of the possibilities of strengthening these resources and using them in times of crisis and especially when the crisis situation becomes chronic [33]. More specifically, social cohesion, social capital as well as trusting community leaders, and community preparedness for emergency and social–communal activities can indeed aid communities’ resilience in times of crisis [34] and in this study’s case, in times of COVID-19. Thus, the present study’s aim is to look at these different dimensions and concepts related to communities and their assets in order to understand their importance and their role during the unique health pandemic of COVID-19. This study broadens the scope of examined Jewish communities by bringing data from various places in Europe, South America, and Central America. Moreover, this study is focused on the mechanisms that communities used to help their members to cope with different aspects of the pandemic experience. Participants were asked to explore challenges, opportunities, and coping during the different waves of the pandemic and following the pandemic. The collected data were then analyzed through the prism of sense of community and resiliency theories.

## 2. Materials and Methods

### 2.1. Participants

A snowball and convenience method of sampling was employed. The author approached several key persons in communities around the world, asking them to participate. In addition, she asked them to request other key personnel and leaders of the community to participate as well. Considerations of the diversity of Jewish communities around the world in addition to COVID-19 restrictions were taken into account. After receiving IRB approval (No. 2021-011), face-to-face, in-person interviews were conducted with leaders and members of the following communities: Budapest, Hungary; Subotica, Serbia; Vienna, Austria; Bratislava, Slovakia; Vilna, Lithuania; Buenos Aires, Rosario, Salta, and Ushuaia in Argentina; and Mexico City and Cancun in Mexico. Descriptions of the role of participants in the communities are described in Table 1.

### 2.2. Interview Procedure

The interviews were conducted between October 2021 and July 2022 and each interview lasted between 45 min and 1.5 h. Some interviews were conducted in Hebrew and others were conducted in English. Oral consent was obtained from each interviewee prior to the beginning of the interview. The interviewees were told that they were being interviewed for a research study and the topic and aims of the study were explained to each of them. With their approval, the interviews were audio-recorded. The recordings were then transcribed so that they could be analyzed and themes could be identified.

The following questions were asked during the semi-structured interviews:Tell me about yourself: name, gender, age, family status, and socioeconomic status.Tell me about the Jewish community that you belong to.What is your role in the community?In your opinion, what role did the Jewish community play for its members during the pandemic?What special activities did the Jewish community organize during the pandemic?What special activities were you in charge of?How were these activities perceived by the community members?What feedback did you receive from the community regarding these activities?What challenges does the Jewish community face today?What opportunities does it have?

### 2.3. Data Analysis

The analysis of this study aimed to address on one hand the research questions and the theoretical basis of the study and, on the other hand, it employed the thematic approach recommended by Lieblich et al. [35]. In the first stage, inductive analysis was applied. That is, a methodology was used in which there is reliance on internal criteria originating from the phenomenon under investigation [36]. According to this approach, each interview is viewed as a holistic unit and researchers then trace the main themes that emerge from the material. The findings were arranged into different categories, according to the discourse that developed during the interview. The second stage included analysis in which a methodological pattern partially focused on criteria, according to which some of the themes resulted from the theoretical framework of the study, in a deductive analysis process. The focus was on the role of the communities during the pandemic, and the challenges, opportunities, and coping that emerged from this event. For each category, there were some subcategories. In the third stage, after identifying the central themes, a map analysis was conducted in which the complete set of categories was created, which included the identification of content relationships between the different units of meaning [36]. The important categories that emerged from the interviews conducted for the present study are presented below.

## 3. Results

Three major themes emerged from the interviews that were conducted around the world: challenges, coping, and opportunities. Each of these themes and their respective subthemes are presented below. Most of the themes were found to be common to the different communities. Although the themes that emerged from the interviews were not only related to COVID-19, this paper focuses only on challenges, coping, and opportunities due to and during the pandemic. Quotes that represent each theme are presented below.

### 3.1. Challenges

Jewish communities around the world were facing various challenges even before COVID-19. In some cases, the pandemic intensified these challenges and, in some cases, it provided some solutions for challenges.

#### 3.1.1. General Challenges

Participant, Vienna, Austria:


*…[L]ike everyone else, the Jewish community was hit by a pandemic, which is not only a medical threat, but an economical and psychological threat. In the beginning, no one knew what it was. Everyone was scared….*


Participant, Salta, Argentina:


*…In a community of 130 families, 30 [people] passed away as result of COVID. Sometimes both grandparents died in the same week; the community had significant losses. Many were sick as well. When we came to celebrate Rosh Hashana [the Jewish New Year] usually people have permanent seats [in the synagogue], it’s a kind of a habit, and now no one sits in the specific seat where X used to sit. It’s very sad.*


#### 3.1.2. Education

Most of the communities included in this study have Jewish schools and those schools, like other schools, faced various academic and social challenges. The main challenges discussed were online learning and the social lives of the pupils.

Participant, Rosario, Argentina:


*It was difficult to organize the Seder Pesach [Jewish ceremony for the Passover holiday] that we usually celebrate in school, on Zoom.*



*Hebrew studies were also a challenge, since not everyone has the Hebrew alphabet on their computer. The number of hours on Zoom was also a challenge.*



*Parents had difficulty paying because their income was reduced; also they thought that what their kids were getting on Zoom was not worthwhile like the regular school they were used to.*



*The kindergarten suffered the most. To have activities over Zoom at that age is almost impossible.*


Participant, Salta, Argentina:


*…[I]t was very difficult because the parents were already tired from the regular school assignments, so to bring on top of that ‘after-school’ Jewish school assignments, it was almost impossible. We tried, but almost 60% didn’t enter the Zoom.*


Participant, Buenos Aires, Argentina:


*2020, everything was closed, no one was allowed to be in the street, all schools were closed, it was forbidden to accept parents who wanted to come to the school. No physical contact.*


Participant, Buenos Aires, Argentina:


*…We knew that the parents would not be able to pay the school; everything was closed. The other schools get support from the government, but for us, it was a long story. We get 40% support, but all the paperwork used to be submitted as hard copies and now everything had to be signed by many people. We had to get approval to sign electronically.*


##### Online Learning

Participant, Budapest, Hungary:


*…We should go back to 2020. So last year, in the middle of March, it was announced that all schools would be closed and there was a lockdown…. Which was really complicated due to the fact that most families of the school are middle-class Jewish families, who could not afford to get all the technical stuff that was needed… Judaism or Hebrew, there were things that cannot be communicated through online teaching; there needs to be a person present. For Hebrew teachers, it was challenge; there were no materials.*


##### Social Situation

Participant, Budapest, Hungary:


*…[S]ome kids got used to it quickly; they were together although they were at home. Another thing, it was difficult, they were lonely….*


#### 3.1.3. Cultural and Social Activities

Participant, Italy:


*Cultural activities were forbidden….*


#### 3.1.4. Social Services

Participant, Budapest, Hungary:


*…[T]he personal contact was needed. We were suffering from not having this contact….*


#### 3.1.5. Health

Participant, Vienna, Austria:


*People that usually went to the community center for it now couldn’t….*


Participant, Vienna, Austria:


*…[T]here was a discussion: Do we need masks? How is the infection being transmitted? Is it by hands, by aerosols? So it affected not only the religious side—prayers, bar mitzvot, celebration, holidays. It was Purim [a Jewish holiday in March], no one really knew what to do….*



*…The biggest challenge was the elderly. Unfortunately, we had several infections…. This wasn’t just a medical challenge; there were also psychological impacts… Families weren’t able to visit…. It was also a challenge to convince everyone to get vaccinated…*


Participant, Budapest, Hungary:


*It was Purim [a Jewish holiday in March], they asked me, ‘What will be?’ I said, ‘Close everything.’ They said, ‘What???’ The museum manager almost had a heart attack. He said, ‘How can we close?’... There are security people in the front…. Who is crazy enough to stand there and check people while there is a pandemic outside and no one knows how you get infected? We knew nothing….*



*In the nursing home, there were incidents of COVID, but not too much death….*


#### 3.1.6. Jewish Life

Participant, Vienna, Austria:


*…[A]t first, everything was closed, so we thought about how we could cater to the community in terms of food for the elderly, how to cater to daily prayer without going to the synagogue. Today, the challenge is to bring people back, to join the tefillot [prayer services] in person, because they join online via Zoom, they think, ‘I didn’t miss so much by not actually going out.’ Also, people are very comfortable and got used to actually staying at home on the couch. So that’s gonna be the biggest challenge, bringing the people back out to be physically united in the community again.*


Participant, Subotica, Serbia:


*The leaders were afraid; they shut the doors [of the synagogue].*


### 3.2. Coping

The different challenges inspired different responses in the different communities. Standard and creative ways of coping are presented below. It should be noted that not all of the ways of coping described here are unique to Jewish communities. However, it seems that communities that had solid institutions and well-known ways of making decisions before the COVID-19 pandemic benefited from that structure and from those institutions during the pandemic. The communities sometimes replaced national or local governments in dealing with and answering their members’ needs.

#### 3.2.1. General Coping

Participant, Vienna, Austria:


*So we immediately coped very well. We immediately set up a crisis-management team, in the first week. Up till now, we have had this management team; its medical doctors, experts from the community; the security department took care of all the measures and how to implement them and there is the political side of it, which is me, the president, and two heads of units from two different departments….*


#### 3.2.2. Coping in the Context of Education

Participant, Budapest, Hungary:


*…We were lucky because one of the principal’s friends from China told her that something bad was going to happen, so that is why she had earlier information, before the official lockdown…. Due to our early information, the school started much earlier to get ready and establish networking….*



*We were very lucky because our teachers who were sick and had to stay home, we had all the technical devices so they could work from home. The teacher was at home, the students were in school…. but they could connect. That was another difficulty, when half of the class was here and the other half was at home—hybrid teaching. The teachers taught in the classroom and at home. This was most difficult, but we tried to do our best.*



*We were also very lucky because, in our school, we have two counselors and they were 100% ready to communicate with the kids and parents. Always ready to help, and we really needed them.*


Participant, Vienna, Austria:


*…[W]e try to [provide] support with laptops, tablets, because we have home-schooling and we try to have psychosocial support on site to assist the parents. Even when there was a lockdown, we provided. We had one class and one kindergarten class that we had open for people who had to go to work because, for example, they are doctors, and they were still able to bring the kids to school, in order for them to go to work.*


Participant, Italy:


*…In the school, it was closed and they organized long-distance learning. They did what they could….*


Participant, Italy:


*[W]e did a lot in the community. In the school, we were the first to try…. We were the first school among the schools in Italy to go to learning long distance. We organized it quickly.*


Participant, Rosario, Argentina:


*…A week after everything was closed, the school was in everyone’s kitchen. We were out of the physical school the entire year of 2020….*


Participant, Salta, Argentina:


*…Then we were allowed to come back with different protocols, for example, classes were only in a big hall and not in classrooms. Now, no one wants to return to the classroom; it’s much more fun, we play, run. I also don’t want to return to the classroom; there is more room this way…. For adult education (40- to 50-year-olds), they decided to stay on Zoom. It is much more convenient for them this way.*


Participant, Buenos Aires, Argentina:


*AMIA [a Jewish organization] gave the school more money and at earlier stages. This support was important, not only economically.*


Participant, Buenos Aires, Argentina:


*The adult Hebrew schools (ulpan) did not stop and even grew… everything was done by Zoom.*


#### 3.2.3. Coping and Cultural and Social Activities

Participant, Italy:


*…[W]e organized video calls and things (in the nursing home)….*



*…[T]he volunteers went to the supermarket and brought [back supplies]. We helped all the vulnerable in the homes, so that they could survive in the best [way]. We made a lot of simple calls to people, to ask if they needed any help, and we organized. For example, when we had vaccinations, we helped all the people who were more than 80 years old. We called everybody to see if they were OK with the vaccination, if they needed help, if they need us to make appointments for them, if they need to be brought to the vaccination [site] and so on. Very simple things, but I used to say that the community was present more than ever. We opened a call center, if someone needed medical advice, etc. We did small things, but everything that we could.*



*…[B]ut there were also many interesting cultural programs and many people participated more than when [such events are held] in real life. If, for example, in real life, when we do things, participation is 50–70 people, in the Zoom cultural activities, we had 200–300 people … [including] Italians that are in Israel or from small communities, so this was really much appreciated because mostly it was on Sunday mornings during the lockdown; everything was closed. It was an interesting way of being together. It was not a cold atmosphere, you could see people you hadn’t seen for many years….*


Participant, Italy:


*…We had programs from history to story problems on what was happening on the web, with the dark web with the antisemitism due to the pandemic, music, books, whatever we could. We were bringing different cultural things on Zoom. Sometimes we had 500 participants on Zoom and many times there were a couple of people, but they were waiting for us, from Israel and from Italy. It was like being at home; we were so happy and we really had wonderful programs. We never stop learning and bringing new ideas to get in touch with the people. We have the European Day for Jewish Culture in Europe once a year and, during the pandemic, we organized one completely on Zoom and it was a success.*


Participant, Italy:


*During the COVID emergency, I have organized a lot of online events, one per week every week, presenting our culture, our history, talking about politics, about Israel, and during these events generally speaking about 300–400 people were present and the majority of them were not Jews.… I organized a Jewish festival, invited a lot people also from Israel…*


Participant, Vienna, Austria:


*…[B]y the human aspect …. being there for one another…. Sticking to each other.… [W]e had up to 100 volunteers at each time who just called people, the entire community from 70 years old, we called everyone, myself, workers, youth, everyone who wanted to help; we just called. ‘How are you doing? The community wants to know if you feel OK. Do you need anything?’ Most of the time [they] didn’t need anything, but were so happy to speak to someone for like 5 min…. In terms of the emotional bonding, I believe COVID actually brought people back together, because you had to focus on the core values and it’s not only Jewish, simply the human aspect of it, being healthy, helping each other, and realizing what is important to your health, your family…. Most of them were just so happy to hear someone. They were sitting at homes for weeks without seeing anyone, without having any contact with the outside world. So it helped a lot, it was the human aspect of it…. So we went and set up video calls for them. We were the first ones [old-age homes] in Vienna that families had the possibility to visit in glass cubes, where at least they were close and could see the parents without risk of infection, things like that….*


Participant, Budapest, Hungary:


*We had 120 volunteers. They brought food and medicines to the elderly. It was a real help. We consulted by Zoom and we also had lectures online and we helped over the phone. We had to learn how to do it and to teach the elderly to use it. We had to encourage them to ask for help and not to be shy….*


Participant, Bratislava, Slovakia:


*…[M]any people became isolated at home, so we started, in March–April, to distribute on our Facebook and in direct mailings speeches of distinguished members of the community, supporting people and encouraging people to stay at home, explaining the situation. We hired a psychologist who lectured and had direct training in isolation… Cultural activities, religious activities on a regular basis, which we realized that if we organize a lecture or a concert or even a minyan [prayer service], pre-pandemic the attendance was much less then online during the pandemic because people were looking to socialize….*


Participant, Salta, Argentina:


*At the end of the year, the government allowed parties, so we did Kabbalat Shabbat [welcoming the Jewish Sabbath on Friday evening] and everyone came, everyone. We wanted to be here.*


Participant, Buenos Aires, Argentina:


*…[W]e did everything online, teachers’ workshops, conferences, all our activities, we did a virtual memorial ceremony [for the victims of the 1994 bombing of a Jewish community center in Buenos Aires]….*


Participant, Buenos Aires, Argentina:


*…[W]e did lectures with different people, psychologists, economists, organizations experts, principals of Jewish studies; 150 joined these lectures and it was very important.*


#### 3.2.4. Coping and Social Services

Participant, Subotica, Serbia:


*This year, there are vaccinations, most members are getting vaccinations, and the Jewish community is giving, one time a week, a free dinner, free lunch; 80 people need the social services that they get from the community. The Holocaust survivors are in a better position now. Children get scholarships for studies.*


Participant, Italy:


*From a financial point of view, we got many donations, small donations from older members of the community, but it was very important, and we also got donations from the Jewish Agency and other Jewish organizations worldwide. Because at the beginning, Italy was in a very bad situation. And the welfare could increase the financial aid and this also helped with volunteers to get food to people.*


Participant, Vienna, Austria:


*…[M]aking donations, everybody was doing what they could, buying food, donating laptops…. We always made a note; they were trained by a psychosocial worker to hear certain signs of loneliness or deterioration…. ‘What’s the purpose of living anymore?’ Immediately they would give it to a psychologist or psychosocial therapist in the center and they did a follow up. So this enabled us to detect some warning signs, because they as professionals could have done it…. with youth organizations, we were shopping. Actually doing the grocery shopping and bringing it to their homes. Cooking meals in the homes of the elderly and bringing meals to their homes.*



*…We tried to give them food and support them. We added to the crisis team a financial-support fund. Up to now, we have put 350,000 euros into it. The municipality and government also tried to do it, but it took months for them. So specifically for the community members and small enterprises, there was a fund for individuals and up to 15,000 people without any bureaucracy. Within one week, they could apply, write what do they need [money] for and how much, and they received it. Another fund was for small enterprises in the community, so in total, we had like 700,000 euros that we distributed in the community for emergency financial help. The infrastructure helped with shopping, etc.*


Participant, Budapest, Hungary:


*The economic situation changed; everything got to be online services. For example, the psychological services. The money that was saved from everything that wasn’t happening because of the pandemic went to people who fell financially because of the pandemic. We changed the plans and succeeded. We were proactive—we supported people financially, we helped with payments to the Jewish school, we helped people who could no longer support their elderly parents. We had focus groups that thought about where the money should go. There wasn’t a real check of who is Jewish; anyone who claimed to be a Jew received the support. Because everything was online, it was quicker. People didn’t feel good about asking for financial aid, but there were the new poor. There was also an option to ask for online psychological services.*


Participant, Bratislava, Slovakia:


*….We have an old-age home…. They distributed meals to the homes of the older people; many of them don’t have families. We have a Social Services department that is very dedicated.*


Participant, Salta, Argentina:


*The board created an emergency group, like a 911 of the community. It still exists today. When someone knew that somebody was in trouble, they called them, they brought them whatever they needed, food, alcogel [hand sanitizer], medicine. For one and a half years, it was very active.*


Participant, Buenos Aires, Argentina:


*When the pandemic started, we brought food to the elderly who used to come here to the Jewish center for lunch and now couldn’t. We also called them, because they needed to speak to someone. There were many volunteers and we could organize this quickly. The community people felt that they were cared for and the employees felt that they were significant for others.*


Participant, Buenos Aires, Argentina:


*The global Jewish Agency for Israel gave big loans to the Jewish communities, part of it became a grant. Nothing that is done by the Jewish community is budgeted by the government. There was a lockdown for 8–10 months. If the Jewish Agency didn’t help, people would not be able to pay for the Jewish schools and other services offered by the community. When your income is being harmed, the first thing you give up is private (Jewish) school for your kids. The communities also had to help with food and medicine for their members, so the Jewish Agency helped the community with funds. The loans were given to the communities and they decided how to distribute [the money]. This was also true for other countries in South America, like Chile, Uruguay, Brazil, but to a lesser extent.*


#### 3.2.5. Coping and Health

Participant, Budapest, Hungary:


*I am always one step ahead of the government. Today, no one has to wear masks, but we require masks in synagogues and you need to have a Green Pass. In the church, you don’t need [a mask]. Hungarians, as opposed to tourists, do not wear masks.*



*If one child was infected, the principal asked for government permission to send the whole class into quarantine. That is how we sent each class into quarantine. Due to this logistic approach, we were able to end the year with the least cases.*



*Every two days, we had a meeting, a decision system; we had several expert leaders of the institutions; everybody gives the information. We were hit hard in the situation, because synagogues. We closed the university. In the first stage, we didn’t have… the mortality was low although the age of the population is high…*



*…The hospital took care of the nursing home and gave them everything they needed. When the vaccine arrived, they also vaccinated them.*


Participant, Subotica, Serbia:


*The leaders were afraid. They shut the doors, be careful, no prayers in the synagogue….*


Participant, Vienna, Austria:


*And even before the government forced us, we decided to take precautionary steps to avoid risk. We limited participants and asked people to stay away and to keep it small. And now, as I look back, it might have been the best decision the president has made…. We wanted to have best practices. We wanted to show that we are at the peak of what we do, in terms of implementing the government restrictions. The government said we need to do quick COVID tests, we’re doing PCR tests. They’re recommending masks, we require masks. They said you don’t have to keep distance in the synagogue; we still kept distance because of pikuach nefesh [the Jewish principle of protecting human life].*



*…[T]he vaccination started, we vaccinated all the Holocaust survivors with [shots given by] Jewish doctors [working] as volunteers, 400 beginning in January. Our institution was based in the old-age homes and, in total, we vaccinated 3000 members, organized all by ourselves. We didn’t have to do it, but for us the biggest role, goal, was to have the best protection for community members…. They could also ask the doctors questions in Hebrew. It was always a collaboration with the municipality and the government. We advocated to get a certain amount of vaccination [doses] and then the cooperation was with the doctor of the city, it was always an official collaboration between the municipality and us and we took responsibility for organizing that and bringing the elderly people from their homes with a driver, picked them up to bring them there.*



*I believe that some of them would not have been vaccinated if it wasn’t for us, because they trust the community, they knew the doctor, they knew the environment in the community center and the old-age home. It wasn’t a random vaccination site in the city.*


Participant, Italy:


*The priority was to defend the health of our old and vulnerable people in the nursing home. We organized new procedures and protocols. The home was closed…. I must say that the focus was on health. And in the first year, we had no cases of COVID in that home….*


Participant, Italy:


*…[A]ll the news we had from Israel, but also from here, from the university. They were proposing, for the first time, a way to do the PCR tests. For our young students, it was fantastic. We organized it.…*


Participant, Bratislava, Slovakia:


*…[S]o from the first days, we purchased FPP2 and 3 masks and distributed them for free among our members. This was March–April 2020, the very first weeks of the pandemic in our region…. We organized antigen testing throughout the last years, I don’t know 5 or 6 times. This we organized together with the municipality, so it was open not only for community members, but also for the general public…. And this year, at the beginning of the year, we organized, with the Ministry of Health, the vaccination of Holocaust survivors and elderly people in the community. This was a unique program, even the minister of health came himself and vaccinated people, so this was a great event…*


Participant, Salta, Argentina:


*We have gotten help from the Joint, alcogel [hand sanitizer], masks, it wasn’t always available….*


#### 3.2.6. Coping and Jewish Life

Participant, Budapest, Hungary:


*…Zoom lectures of the rabbis started and it was comfortable….*


Participant, Subotica, Serbia:


*…[T]here was Rosh Hashanah, Yom Kippur [Jewish holidays in the fall], there were classes, some prayer; there were never more than 6–8 men at the same time in the Jewish ceremonies….*


Participant, Vienna, Austria:


*…Virtual prayers, except for Shabbat [the Sabbath], for example.… As long as the supermarket is open for me, religion and the tefilla [prayer] are as important and can be open as well. So, I tried to find compromises for them, [places] where it would be healthier. So we would do things like outdoor prayers in backyards. We would provide them with disinfectant spray, masks… We would give them quick corona tests for free. We would give them masks, disinfectant spray and ask them to use them, also tell them, ‘Listen, we want to [use the] best practice, it’s gonna be a chillul hashem [an embarrassment for the community] if you do this…’*


Participant, Venice, Italy:


*[T]here was Zoom (Jewish) education, Zoom Purim, the only holiday that it [use of electronics] doesn’t conflict with the halacha [Jewish law], the rabbis teaching on Zoom and YouTube.*


Participant, Salta, Argentina:


*There were social online activities in Argentina, so we got connected, prayers, courses, help, etc.… There was also a psychologist group that volunteered; everyone could choose to speak to someone and it was very successful. It was from our Jewish community. There was a list and we could choose who we wanted to speak to. It was very important, because we were very anxious.*


### 3.3. Opportunities

#### 3.3.1. Opportunities for Social/Jewish Life

Participant, Budapest, Hungary:


*I am more optimistic; 700 families that had never before made contact made contact during COVID. If we know how to take advantage of this momentum, maybe we will be able to achieve something….*



*In the past, we built all the institutions, but today, it’s the members, the community members that are most important and I think that, due to the crisis, we had the opportunity to work with the people in the community, to give them money and social connection, to put the emphasis on them and not to invest in stones. I hope that next week, I’ll have a meeting with all the important people and I want to change attitudes, so that we focus on community members. Now, the community is weak, but it is an opportunity for great improvement.*


Participant, Bratislava, Slovakia:


*…[T]his how we came to the idea of establishing the TV platform [Tachles TV] which is now developed as a Czechoslovak platform, to attract people from both countries, which are culturally and linguistically related, and this is one of the best outputs from the pandemic.*


Participant, Salta, Argentina:


*We found out that it’s easier to stay in contact through the internet, so now we can be in touch with Israel and Israelis who were born here, but don’t live here anymore. We can be together and share different things. Since we are a very small community and there are even smaller communities they can join us via the internet for different online activities. For example, we still study Judaism with the Tuckuman rabbi. The north of Argentina is united for special events, such as that.*


Participant, Buenos Aires, Argentina:


*There was a group of youngsters, 18–25, sitting at home with nothing to do. So we organized the Masa project for them, where they went to Israel, to Tel Aviv, and got the experience of being ‘Israelis’... They also benefited from getting vaccinated, since here it wasn’t available.*



*People understood that there is no future for them here. They didn’t see their future here. For some, their kids or parents are living in Israel and COVID has separated them for long periods. So they understood that they want to immigrate to Israel and for us [the Jewish Agency], that’s the goal. It’s a great success.*


#### 3.3.2. Educational Opportunities

Participant, Budapest, Hungary:


*…In the Jewish university, we have an opportunity to arrange the study programs now. We received from the government 25 million euros, maybe we will be able to bring in more students. Actually, those who are from outside of Budapest do not really have to come; we can still continue the online learning. But, if we continue to take advantage of this, maybe it will also strengthen the community. We don’t really know yet.*



*…We were very strongly supported when the pandemic started by the Jewish community, our sponsor, and, therefore, the first teachers in the country to be tested were ours. All of us were tested; five teachers were taken out of the school immediately. We were grateful to the hospital and for the community, for being tested. All the other schools and teachers were tested much later.*


Participant, Rosario, Argentina:


*All processes were quicker. What we used to do in a few weeks now turned to hours, because we had to give answers all the time…. Because we functioned so well, parents from the public school who were unhappy there have moved their kids to our schools, because they felt it was better. So, now, we also have non-Jews in our school, but they study Hebrew, Bible, and accept Zionism as part of the school’s curriculum.*


Participant, Salta, Argentina:


*We have developed a program called To Bloom, to which each teacher contributed one or two hours of work, and we built a virtual school, for those who do not have a community. A year ago, we reached about 70–80 students. Various ages, interesting curriculum, Hebrew at several levels, arts, Bible, each person has entered whatever they like. It is very special.*


Participant, Buenos Aires, Argentina:


*…All the teachers and principals from around the country could join meetings together and also help plan the year of 2021. If, before, not everyone was coming to these meetings, now it is easier…. We joined different groups that offer training. So, we can join others’ training [sessions], so now the amount of courses that are offered is much bigger and wider. Most teachers are very happy and thank us for this.*


Participant, Buenos Aires, Argentina:


*People were staying and sitting at home, so we understood that there is an opportunity here; the Hebrew schools for adults doubled themselves. It gave people something to do; it gave them hope.*


#### 3.3.3. Opportunities Related to Health

Participant, Budapest, Hungary:


*…When there was a need for vaccinations, we gave them. We vaccinated thousands; we didn’t check if they were real Jews. If someone told us they were a Jew, for us, they were a Jew. If a person tells us that he is a Holocaust survivor, I accept it. The Meraj Association works with the Germans; it gives help and money, food and social services. Two thousand people use it. All the emergency services are now run through the hospital. If there is someone who we think has COVID, we can immediately run a PCR test.*


## 4. Discussion

Based on the sense of community theory [1,2], resilience theory, and related concepts, the aim of this study was to examine the ways in which Jewish communities around the world helped their community members to overcome the lockdowns and other difficulties that emerged as a result of the COVID-19 pandemic. By exploring challenges and opportunities that the pandemic brought to the world, we were able to pinpoint the role of these communities in their members’ physical and mental health.

In this discussion, we will deepen our understanding of the ways in which the different concepts of theories of community, namely, sense of community and community resilience, served members of this minority group around the world. The different concepts of these theories are closely related to each other. Therefore, it was quite hard to make an artificial separation and some ways of coping could fit more than one concept. However, for the sake of this discussion and summary, I chose the concept that fit best.

### 4.1. Community Preparedness

The interviews with individuals from the various Jewish communities who took part in this study revealed that those communities that had solid institutions and well-known ways of making decisions prior to the pandemic were able to utilize and benefit from those structures and institutions during the pandemic. It was relatively easy for them to adapt to the situation and make important decisions for the sake of the members of their community. Because of their adeptness, they sometimes replaced national governments or municipalities in dealing with and answering their members’ needs. Their competence and effectiveness brought even people that did not belong to the community to partake in what the Jewish communities were offering their members. For example, the quick and effective way in which Jewish schools responded allowed them to function well quickly. This led some additional Jewish parents and even non-Jewish parents to enroll their children in these Jewish schools, because of the benefits offered and the abilities of those schools to function better than other local schools. Another example is vaccinations, which were more quickly brought to some Jewish community members, as well as other elders in need.

### 4.2. Membership, Belonging, and Social Ties

Being a member of a Jewish community allowed individuals to experience a sense of belonging to the community during the crisis, which allowed social ties to be maintained in various ways. It seems that the leaders of the communities acted to the best of their abilities to maintain the sense of belonging and the social ties of community members who could no longer meet in person. For example, the communities created unique cultural programs and activities that were delivered online. The organizers reported that similar, pre-pandemic in-person programs were successful, but not many people participated. However, during COVID-19, hundreds of members participated in each activity. Since they saw the need for people to get together, they also created purely social online gatherings. Being a member of a Jewish community brought other benefits in different places around the world. For example, in Bratislava, they created a TV channel (Tachles TV) with programs tailored to Jewish communities and their members.

### 4.3. Social Support and Emotional Connections

Social support in the context of community membership is a form of emotional, as well as material and instrumental, assistance that individuals receive from other members of their community. This could be on an everyday basis, but the support usually accelerates and is manifested especially in times of crisis. Indeed, during COVID-19, the Jewish communities included in this study put this value at the forefront of their activities in many ways. First, elderly people were supported from the very beginning of the pandemic in many communities. Volunteers were organized to shop for groceries and other basic needs of the elderly. Moreover, volunteers also made telephone calls to elderly community members, to relieve loneliness and to check how they were doing. Counselors and psychologists were recruited to telephone vulnerable individuals (not only the elderly) to provide support. As for Jewish life, in order to maintain an emotional connection, Jewish services, prayers, and holidays were shared in online Zoom meetings. All of these activities helped community members to feel emotionally connected to each other and, in this way, social support was created.

### 4.4. Integration and the Fulfillment of Needs

Integration and the fulfillment of needs can strengthen community members’ feelings of togetherness and their sense of obligation toward the community. Indeed, it seems that community leaders did everything that they could to fulfill the needs of the members of their communities. The economic burden faced by community members led them to raise funds, to support members who were unable to support themselves. Fundraising was an important, meaningful action of the leaders and this fundraising allowed them to preserve community institutions and to continue community activities. For example, several communities supported parents by waiving school fees and others distributed meals to those in need.

In terms of health, the leaders of the various communities included in this study kept strict rules and did not allow in-person meetings at times when they were not legally allowed. They moved Jewish services online, so that people could feel that their religious needs were being supported. Moreover, when the time came to meet again in person, they were careful to distribute masks, COVID-19 tests, and hand sanitizer. In addition, when vaccinations arrived, they took care of the elderly by bringing them to be vaccinated. They explained that some community members would not have gotten vaccinated, if the vaccinations had not been administered by a Jewish doctor whom they knew and trusted.

### 4.5. Study Limitations

This study included a number of different Jewish communities from Europe and from Central and South America, which were selected to reflect different types of communities in different parts of the world. Therefore, the results should be carefully viewed and readers should be careful about drawing conclusions to other communities around the world. Also, most of the interviews were with community leaders and others who held prominent positions in the community, as opposed to other members of the communities. Other members of these communities might have focused on different topics and drawn different pictures of coping during the pandemic in their communities. Despite these limitations, the importance of this study lies in the fact that it is field research carried out during the stressful COVID-19 pandemic. Future studies should explore the views of members of other Jewish communities and other minority communities around the world.

## 5. Conclusions

Beyond giving voice to an understudied group of Jewish communities, as minority communities, during the COVID-19 pandemic, this study teaches us about the challenges and opportunities encountered by the leaders of Jewish communities during the stressful situation of a global pandemic. Our results show that strong communities can benefit their members and take care of them during such a harsh time. The ways in which leaders handle a crisis affect the members of their communities. It seems that despite the fact the Jews are a minority group in most of the countries in which they live, their strong communities were an asset during the pandemic and took care of them very well, in terms of physical health, mental health, and quality of life. Other communities could learn from the practices of the communities examined in the present study, as they prepare for future crises.

## Figures and Tables

**Table 1 ijerph-20-01107-t001:** List of the interviewees.

Participant	City, Country	Role in the Community
1	Budapest, Hungary	President, MAZSIHISZ—Federation of Jewish Communities in Hungary
2	Budapest, Hungary	Head of the Yahad community service, MAZSIHISZ—Federation of Jewish Communities in Hungary
3	Budapest, Hungary	Adjunct professor, deputy dean, ORZSE—Jewish Theological Seminary, University of Jewish Studies
4	Budapest, Hungary	Secretary of foreign relations of the Jewish community
5	Budapest, Hungary	Principal, Jewish school
6	Budapest, Hungary	Manager, Jewish hospital
7	Subotica, Serbia	Former president of the community
8	Subotica, Serbia	Former director of the community
9	Subotica, Serbia	Former leader of the community
10	Bratislava, Slovakia	President of the Bratislava Jewish community
11	Vienna, Austria	Secretary-general of the community
12	Milan, Italy	Vice-president and councilor for welfare and old-age homes
13	Milan, Italy	Councilor for culture
14	Milan, Italy	Deputy councilor for culture and schools
15	Venice, Italy	Former community leader
16	Israel	Co-president, Mizrachi World Movement
17	Buenos Aires, Argentina	Former Jewish Agency emissary to Argentina
18	Bariloche, Argentina	Rabbi
19	Buenos Aires, Argentina	Secretary-general of the federation of Jewish schools in Argentina
20	Buenos Aires, Argentina	CEO AMIA—Jewish Community of Argentina
21	Buenos Aires, Argentina	Director, Central Jewish Education
22	Buenos Aires, Argentina	Director of the members’ department, AMIA—Jewish Community of Argentina
23	Buenos Aires, Argentina	Manager, Jewish Agency, promotion of immigration to Israel from Spanish-speaking countries
24	Buenos Aires, Argentina	Jewish Agency emissary to Spanish-speaking countries
25	Buenos Aires, Argentina	CEO Joint—South America
26	Ushuaia, Argentina	Hebrew teacher
27	Salta, Argentina	Hebrew teacher and principal of the Jewish school
28	Rosario, Argentina	Principal, Jewish school
29	Vilna, Lithuania	President of the Jewish community
30	Cancun, Mexico	Former president of the community
31	Cancun, Mexico	Secretary of the community
32	Mexico City, Mexico	Researcher, Jewish University

## Data Availability

Data collected and used in this study are available from the author upon reasonable request.

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
