# Peer review of "Challenges, Opportunities, and Coping in the Wake of the COVID-19 Pandemic: The Case of Jewish Communities around the World"

_ijerph, 2023, doi:10.3390/ijerph20021107_

Round 1
Reviewer 1 Report
The findings of this study as described are significant and rich in nature. However, in my opinion, their presentation and authenticity could be improved in several ways:
1) The introduction is comprised largely of a review of literature on key aspects and dimensions of 'community'. While this is commendably thorough and comprehensive, it is not much linked to the study that follows and related to this it does not substantiate the case for the study's focus on Jewish communities, intrinsically justified as this may be. It would be helpful for the introduction to be more systematically structured so as to relate clearly to the study that follows.
2. The piece does not foreground with any clarity the definition(s) of 'community' on which the study is based; though this is clarified to some extent in the subsequent content that seems to speak of 'community' as defined exclusively by population level cultural and religious parameters - again, justifiable and important as this obviously is. Related to this, there are some contradictions in the basis for the study's conclusions and the relationship of the conclusions to the ways in which the basis for the study is described. For example, the categorical statement that defining feature of a community's ability to cope with crisis (line 109), and the presentation sits uneasily with the following sentence which suggests that the nature of community resilience and wellness are multi factorial. While the multi-dimensional character of the concepts are acknowledged, the lit review sections presents them more as definitionally irrefutable.
3. The methods section is thin. It does not describe the criteria for selecting particular communities or countries or for how the participants from within each community were selected. Most participants appear to have been figures of some authority within their communities (I am not sure why they their roles were not identified in such as a way as to have linked them to responses since this would have added interest and relevance to the findings) but importantly there seems to be no obvious reason for not including the perspectives of more 'ordinary community members. The interviews asked the question: ' tell me about the Jewish community you belong to' but no previous explanation was provided as to the relationship of the sites chosen and therefore the scope and implications of the answers to this question.
The data analysis is thinly described, particularly in relation to the process and criteria of theme selection. For example, what, in the context of this study are the defining parameters of 'a holistic unit'; what does this mean?
Author Response
Attached, please find a detail description of the changes.

Reviewer 2 Report
Overall, I find the topic compelling and appreciate the international scope of the data collection. However, I have concerns about the theoretical foundations of the research, as well as some of the methodological and analytical decisions.
With regards to the theoretical foundations, the author mentions "sense of community", "community resilience", "social capital", and "social cohesion", but does not attempt to disentangle these concepts from each other. There is a lot of theoretical and conceptual overlap that requires a bit of work to make clear why they are essential for this particular study.
With regards to the methodological approach, I would have liked to have seen more explicit reflection on why a qualitative approach was taken, and specifically, why thematic analysis was chosen, and exactly how it was carried out.
Relatedly, a number of times, it was stated that three themes "emerged" from the data, indicating a (somewhat passive) inductive approach, namely: challenges, coping, and opportunities. But other times (e.g., Lines 39-40) the text described a more deductive approach that implied that the researcher was rather searching for these three themes in the data.
Moreover, it is not clear why the author presents a series of quotes from the interviews within these three "themes" in section 3, but offers little to no analysis of these quotes. The paper would be strengthened tremendously by being more selective when choosing verbatim quotes and instead making the foundation of the results more analytical. For example, it seems that the information in sections 4.1-4.4 would actually be more useful in section 3, using the quotes to help illustrate these points.
Although that being said, I wonder if it is really necessary to force the data to fit into the preconceived concepts of sections 4.1.-4.4. If the author is truly aiming for an inductive approach, they would create themes that are grounded in the data. Then the discussion could include what these data add above and beyond what has been found in previous literature. In fact, there is no attempt made at relating the findings to previous literature, which makes it difficult to judge the contribution that the current paper is making.
Altogether, the paper is lacking in clarity and would benefit from a fundamental revision.
Author Response
Thank you very much for giving us the opportunity to revise and resubmit our article. You will find our revised manuscript attached. I am grateful for the valuable comments, which I believe improved the quality of the manuscript. All changes were saved in ‘track changes’ mode. The following are our specific responses to each comment.
Reviewer 2
Overall, I find the topic compelling and appreciate the international scope of the data collection. However, I have concerns about the theoretical foundations of the research, as well as some of the methodological and analytical decisions.
With regards to the theoretical foundations, the author mentions "sense of community", "community resilience", "social capital", and "social cohesion", but does not attempt to disentangle these concepts from each other. There is a lot of theoretical and conceptual overlap that requires a bit of work to make clear why they are essential for this particular study.
The introduction indeed focuses on the sense community theory and the unique characteristics of communities that aid their member to cope well during the crisis and for this study the Covid-19 crisis. In order to clear this point, in the present version of the manuscript I have added a paragraph that specifies the importance of minority communities and the features of communities in helping their members to cope with crises. (See p. 1).
Additionally, to address the above comments, I have added a paragraph on p. 5 under the section “The current study”. In this section, I have tried to clarify the reasons for reviewing the various definitions throughout the literature reviews. This was done since these different aspects and elements all have potential preventive characteristics to aid minority members of the communities in overcoming stressful events especially when it becomes chronic like during the pandemic. I hope that this elaboration is giving an answer to what has been seen a bit scattered. (See p. 5).
With regards to the methodological approach, I would have liked to have seen more explicit reflection on why a qualitative approach was taken, and specifically, why thematic analysis was chosen, and exactly how it was carried out.
I agree that the previous method section was rather thin. Therefore, I have added an explanation for the method of sampling which was approached. In addition, I have also added the different stages of the analyses. Thus, the present version illuminates the three stages of analyses which included both inductive and deductive approaches. (See p. 7)
Relatedly, a number of times, it was stated that three themes "emerged" from the data, indicating a (somewhat passive) inductive approach, namely: challenges, coping, and opportunities. But other times (e.g., Lines 39-40) the text described a more deductive approach that implied that the researcher was rather searching for these three themes in the data.
Furthermore, to address this comment, in the present version of the manuscript, I have elaborated on the procedure related to the methods of this study. In the procedure, there are several stages, including inductive and deductive approaches to analyzing the data. Therefore, after explaining the different stages, it seems that being sometimes passive and sometimes more active in analyzing the data fits the different stages of the method of this study. (P. 7).
Moreover, it is not clear why the author presents a series of quotes from the interviews within these three "themes" in section 3, but offers little to no analysis of these quotes. The paper would be strengthened tremendously by being more selective when choosing verbatim quotes and instead making the foundation of the results more analytical. For example, it seems that the information in sections 4.1-4.4 would actually be more useful in section 3, using the quotes to help illustrate these points.
Although that being said, I wonder if it is really necessary to force the data to fit into the preconceived concepts of sections 4.1.-4.4. If the author is truly aiming for an inductive approach, they would create themes that are grounded in the data. Then the discussion could include what these data add above and beyond what has been found in previous literature. In fact, there is no attempt made at relating the findings to previous literature, which makes it difficult to judge the contribution that the current paper is making.
In the present study and manuscript I have chosen to follow the guidelines of Shkedi (2007) – “Words of Meaning – Qualitative Research – Theory and Practice” for writing the report/article. Therefore, I used different sections as suggested in his book. The discussion and conclusion appear according to his suggestion in separate sections and not as part of the results section. I realize that there are different methods of writing a qualitative study report, and should the reviewer require changing the methods of the report I will consider doing so.
Altogether, the paper is lacking in clarity and would benefit from a fundamental revision.
To summarize, I have made changes in the article to take the different comments into account to the best of my understanding. I trust that these changes answer the major (and fruitful) criticisms made by the reviewers and hope that in this revised form the paper is acceptable. I look forward to your reply.
Sincerely,
The author
